# The Role of Adipokines in the Control of Pituitary Functions

**DOI:** 10.3390/ani14020353

**Published:** 2024-01-22

**Authors:** Barbara Kaminska, Beata Kurowicka, Marta Kiezun, Kamil Dobrzyn, Katarzyna Kisielewska, Marlena Gudelska, Grzegorz Kopij, Karolina Szymanska, Barbara Zarzecka, Oguzhan Koker, Ewa Zaobidna, Nina Smolinska, Tadeusz Kaminski

**Affiliations:** 1Department of Animal Anatomy and Physiology, Faculty of Biology and Biotechnology, University of Warmia and Mazury in Olsztyn, 10-719 Olsztyn, Poland; barbara.kaminska@uwm.edu.pl (B.K.); beata.kurowicka@uwm.edu.pl (B.K.); marta.kiezun@uwm.edu.pl (M.K.); grzegorz.kopij@uwm.edu.pl (G.K.); k.szymanska@uwm.edu.pl (K.S.); barbara.zarzecka@student.uwm.edu.pl (B.Z.); oguzhan.koker@student.uwm.edu.pl (O.K.); nina.smolinska@uwm.edu.pl (N.S.); 2Department of Zoology, Faculty of Biology and Biotechnology, University of Warmia and Mazury in Olsztyn, 10-719 Olsztyn, Poland; kamil.dobrzyn@uwm.edu.pl; 3Department of Human Histology and Embryology, School of Medicine, Collegium Medicum, University of Warmia and Mazury in Olsztyn, 10-082 Olsztyn, Poland; katarzyna.kisielewska@uwm.edu.pl (K.K.); marlena.gudelska@uwm.edu.pl (M.G.); 4Department of Biochemistry, Faculty of Biology and Biotechnology, University of Warmia and Mazury in Olsztyn, 10-719 Olsztyn, Poland; ewa.zaobidna@uwm.edu.pl

**Keywords:** pituitary, adipokines, reproduction

## Abstract

**Simple Summary:**

The pituitary gland is a key endocrine gland in all classes of vertebrates, including mammals, and it is an important part of hypothalamus–pituitary–target organ hormonal regulatory axes. In response to hypothalamic stimuli, the pituitary gland secretes a number of hormones involved in the regulation of metabolism, stress reactions and environmental adaptation, growth and development, as well as reproductive processes and lactation. The pituitary gland also responds to a complex of other peripheral signals, including adipose-tissue-derived factors. These substances are a broad group of peptides known as adipocytokines or adipokines that act as endocrine hormones mainly involved in energy homeostasis. Adipokines and their receptors are expressed in many tissues, including the pituitary gland, suggesting that they have a direct effect on this gland. This review is an overview of the existing knowledge of the relationship between chosen adipose-derived factors and endocrine functions of the pituitary gland, with an emphasis on the pituitary control of reproductive processes.

**Abstract:**

The pituitary gland is a key endocrine gland in all classes of vertebrates, including mammals. The pituitary gland is an important component of hypothalamus–pituitary–target organ hormonal regulatory axes and forms a functional link between the nervous system and the endocrine system. In response to hypothalamic stimuli, the pituitary gland secretes a number of hormones involved in the regulation of metabolism, stress reactions and environmental adaptation, growth and development, as well as reproductive processes and lactation. In turn, hormones secreted by target organs at the lowest levels of the hormonal regulatory axes regulate the functions of the pituitary gland in the process of hormonal feedback. The pituitary also responds to other peripheral signals, including adipose-tissue-derived factors. These substances are a broad group of peptides known as adipocytokines or adipokines that act as endocrine hormones mainly involved in energy homeostasis. Adipokines, including adiponectin, resistin, apelin, chemerin, visfatin, and irisin, are also expressed in the pituitary gland, and they influence the secretory functions of this gland. This review is an overview of the existing knowledge of the relationship between chosen adipose-derived factors and endocrine functions of the pituitary gland, with an emphasis on the pituitary control of reproductive processes.

## 1. Introduction

Reproductive success in animals depends largely on the physiological mechanisms controlling hormonal and metabolic homeostasis. Studies in recent years have paid a lot of attention to novel endocrine factors affecting the proper course of the oestrous cycle, implantation, pregnancy, and reproductive outcomes, especially in farm animals. Furthermore, it is postulated that hormones and cytokines involved in energy balance maintenance are also important players in the modulation of female reproduction [1]. White adipose tissue (WAT), besides being a place of energy storage, seems to be the biggest source of biologically active compounds, including so-called adipokines. This huge family of hormones, exerting pleiotropic effects on various tissues and organs, is proposed to form a hormonal link between energy metabolism, inflammatory response, and female reproductive processes. Although most studies focus on the role of adipokines in the control of ovary function [2], their influence on the pituitary gland is also of importance. With the exception of leptin, the influence of which on pituitary functions seems to be relatively well known (approximately 200 publications in the past 5 years), the role of newly discovered adipokines in the control of pituitary functions is known to a much lesser extent.

The pituitary is a small endocrine gland with a key role in the regulation of hypothalamus–pituitary–target gland axes. It forms a physical link between neural stimuli and the endocrine system. The gland secretes several trophic hormones, which are involved in the regulation of metabolism, stress reactions and environmental adaptation, growth and development, as well as reproductive processes and lactation. Besides being a source of hormones, the pituitary gland is also sensitive and responds to many various peripheral signals that modulate its functioning. This study focuses on the role of recently discovered adipokines, such as adiponectin, resistin, apelin, chemerin, visfatin, and irisin, in the control of the anterior pituitary function with a particular emphasis on the functions of this gland in the control of reproductive physiology.

## 2. Adiponectin

Adiponectin (ADIPOQ, ACRP30, apM1) was first described in the mid-1990s [3,4,5,6]. In 1995, Schrerer et al. [3] depicted a protein produced by adipocytes as the adipocyte complement-regulated protein of 30 kDa (ACRP30), the secretion of which increased under the influence of insulin. The concentration of adiponectin in plasma, reaching several dozen µg/mL, is many times (1000 times) higher than that of other peptide hormones and is inversely associated with body fat distribution [3,7]. The primary product of the adiponectin gene (*AdipoQ*), which is adiponectin (30 kDa in its full length), undergoes proteolytic cleavage to globular adiponectin, which can assemble into trimeric forms and higher-order structures [8,9]. In plasma, adiponectin occurs primarily in homomultimeric forms: trimers (low molecular weight), hexamers (medium molecular weight), and various high-molecular-weight multimers composed of 12 to 18 adiponectin molecules [10,11]. Initially, it was believed that the source of adiponectin was exclusively differentiating adipocytes [3]. Subsequent studies conducted in humans, rodents, pigs, and birds have shown that adiponectin is also synthesised by other cells and tissues, e.g., the skeletal muscles [12,13], hypothalamus, pituitary gland [13,14,15,16,17], cardiomyocytes [18], osteoblasts [19], gonads [13,15,20,21], uterus and placenta [22,23,24,25,26], and adrenal glands and liver [13,27]. Adiponectin has pleiotropic effects. This adipokine can control energy homeostasis, insulin sensitivity, and feeding behaviour, and it plays an important role in the regulation of autonomic and neuroendocrine functions [28,29,30,31,32]. Adiponectin exerts its biological effects mainly through two types of specific receptors, ADIPOR1 and ADIPOR2 (encoded by two distinct genes, *ADIPOR1* and *ADIPOR2*, respectively), which were identified and characterised in 2003 by Yamauchi’s team [33]. *ADIPOR1*’s gene expression is high in the skeletal muscle, whereas the *ADIPOR2* transcript is predominantly expressed in the liver. In subsequent experiments, high expression levels for both genes were also found in the pancreatic beta cells [34]. It is currently known that adiponectin receptors are universally expressed in human and animal bodies. The presence of ADIPOR1 and ADIPOR2 proteins or the mRNAs of their genes were found, among others, in the adipose tissue [35], central nervous system [14,16,36,37], reproductive system [15,20,22,24,25,26], heart, lungs, liver, kidneys [35,38], and endocrine glands [34,39]. The effects that adiponectin exerts through ADIPOR1 and ADIPOR2 are different. It has been shown that in mice, the action of adiponectin through ADIPOR1 appears to reduce glucose tolerance, motor activity, and energy expenditure and promotes an increase in adiposity. In turn, ADIPOR2 activation increases glucose tolerance, motor activity, and energy expenditure, reduces plasma cholesterol levels, and increases resistance to high-fat-diet-induced obesity [40]. In 2004, the extracellular protein T-cadherin was identified as the third adiponectin receptor, which may act as a coreceptor binding hexameric and high-molecular-weight adiponectin multimers [41]. This receptor most likely functions as an adiponectin-binding protein and does not directly participate in adiponectin intracellular signalling. It seems that T-cadherin/adiponectin interactions are of great importance in vascular homeostasis and cardioprotection [42,43].

In the pituitary gland, the expression of genes and proteins of the adiponectin system components (adiponectin, ADIPOR1, and ADIPOR2) has been examined in humans [14], pigs [17,37], rodents [32,39,44], and birds [13,15,45]. In humans, a high expression of adiponectin system proteins was observed in the anterior pituitary (in GH-, FSH-, LH-, and TSH-producing cells). Adiponectin and its receptors’ proteins were not colocalised with PRL- and ACTH-immunoreactive cells. In the intermediate lobe of the pituitary gland, adiponectin was detected in gonadotrophs (LH- and FSH-positive cells) and thyrotrophs, but neither the presence of ADIPOR1 nor ADIPOR2 was detected [14]. In turn, in pigs, the expressions of genes and proteins of adiponectin and its receptors were found in both the anterior and posterior lobes of the pituitary gland [17,37]. The expressions of proteins and genes of the adiponectin system components in the pituitary gland are related, among others, to the reproductive status of animals. In the beaver pituitary, the expressions of *ADIPOR1* and *ADIPOR2* were higher in males in relation to females and were the lowest during the reproductive season [39]. The expressions of genes and proteins of the adiponectin system in the pig pituitary changed during the oestrous cycle and were generally higher during the luteal phase than the follicular phase [17,37]. It has been shown that adiponectin can directly influence the secretion of pituitary hormones, although the results of these studies are ambiguous. In rat pituitary cell cultures, adiponectin inhibited GH and LH release [32]. On the other hand, it was observed that adiponectin stimulated GH secretion from isolated rat somatotroph cells [46]. Adiponectin did not affect basal LH secretion but increased basal FSH release by isolated porcine anterior pituitary cells [17]. In the murine AtT-20 pituitary corticotroph cell line and rat pituitary corticotroph cells, adiponectin stimulated basal ACTH secretion [47]. In turn, the pituitary cells isolated from macaques and baboons, under the influence of adiponectin, decreased ACTH and GH secretion and increased PRL secretion, with no changes in gonadotrophin and TSH release [48].

## 3. Resistin

Resistin, also known as FIZZ3 (found in Inflammatory Zone 3) [49] or adipose tissue-specific secretory factor (ADSF), is encoded by the *RETN* gene, and was first described in the 2000–2001 period by three independent groups of researchers [49,50,51]. Resistin belongs to a group of cysteine-rich proteins generally referred to as resistin-like molecules (RELMs), four of which have been identified in mice (resistin, RELMα, RELMβ, and RELMγ). In humans, two proteins of the RELM family are known: resistin and RELMγ [52]. There is a fairly low similarity between human and mouse resistin (59%), and both proteins also differ in the main place of production; in mice, it is produced in the white adipose tissue, while in humans, the main sites of resistin synthesis are peripheral mononuclear blood cells, macrophages, and bone marrow [53].

Resistin is a secretory peptide that interacts with cells through a membrane receptor, although none of the postulated resistin receptors are specific for this adipokine. It has been suggested that these may be Toll-like4 receptors (TLR4) acting through the MAPK pathway, and the phosphorylation of ERK, P38, and JNK lead to an increase in the expression of NF-κB [54,55,56,57]. Other signalling pathways associated with the TLR4 receptor are the PI3kinase/Akt—NF-κB pathway and the AMP-activated kinase (AMPK) pathway [58]. Another postulated receptor is an isoform of the extracellular matrix protein—delta-decorin—which is formed as a result of the proteolysis of decorin. The binding of resistin to delta-decorin activates the signalling cascade of protein kinase A and cyclic AMP, leading to the activation of the pro-inflammatory NF-κB transcription factor, as was found in murine progenitor adipocytes [59]. In mouse adipocyte progenitor cells, another putative receptor for resistin was found—receptor tyrosine kinase-like orphan receptor 1 (ROR1)—with the non-canonical WNT pathway with the WNT5a protein as an activating factor leading to the inhibition of ROR1 tyrosine phosphorylation, the modulation of the MAP kinase pathway, as well as Glut4 and Glut1 expression in 3T3-L1 cells [60]. In rodents, resistin inhibits AMP-activated kinase (AMPK) and induces an anti-inflammatory mediator suppressor of cytokine signalling-3 (SOCS-3). SOCS-3 may mediate resistin-induced insulin resistance and cytokine production, as it is a factor that reduces the insulin response of adipocytes [61].

Resistin was named due to its ability to block insulin and, consequently, impair glucose homeostasis in rodents [50]. As a result of obesity in mouse experimental models, the production of resistin in the adipose tissue increases. Silencing the resistin gene in mice alleviates symptoms of metabolic syndrome, such as hepatic steatosis, increased serum cholesterol, and very low-density lipoprotein levels. On the other hand, in mice expressing human resistin, glucose tolerance and hepatic insulin resistance have been demonstrated under chronic inflammatory conditions, as well as an increased production of pro-inflammatory cytokines (TNF-α, IL-1, and MCP-1) [62]. The pro-inflammatory effects of human resistin have been confirmed in other tissues, suggesting its association with diseases such as type 2 diabetes, rheumatoid arthritis, chronic kidney disease, sepsis, and coronary atherosclerosis [63,64]. The main effort in the study of resistin action is concentrated on its peripheral influences on the metabolic and inflammatory state of different organs. However, the influence of resistin on metabolic homeostasis mechanisms cannot be excluded through its influence on the activity of metabolic control centres in the hypothalamus and pituitary gland, as the presence of resistin and the possibility of its synthesis have been found in rodents [65,66]. The immunolocalisation of the protein showed its highest presence in the arcuate nucleus of the hypothalamus and the anterior and intermediate pituitary gland. In the pituitary gland, resistin expression has been shown to be dependent on the arcuate nucleus of the hypothalamus and changes with age. The destruction of the neurons of the arcuate nucleus significantly diminishes the expression of resistin in the pituitary gland [65]. In the hypothalamus, a shorter form of resistin (s-resistin), which is the intracellular form and is not secreted, was found [67]. The inhibition of s-resistin synthesis resulted in an increase in the activity of leptin signalling pathways and an insulin pathway in the rat hypothalamus. As a result, improvements in glycemia and insulin sensitivity and a decrease in inflammatory parameters were found in rats. Resistin administered intraventricularly or directly into the hypothalamus causes an increase in blood glucose levels, liver insulin resistance, and the production of cytokines TNF-α, IL-6, and SOCS-3 [68].

An increase in the secretion of GH and ACTH by resistin has been demonstrated in vitro in cultures of primary cells of the anterior pituitary gland of macaques and baboons [48] and a somatotroph cell line [69], and this effect is caused by intracellular signalling pathways similar to GHRH. These findings further strengthen the involvement of the hypothalamo-pituitary system in the development of the metabolic syndrome. The inhibition of LH secretion by resistin was found in the mouse cell line of LβT2 gonadotrophs by increasing the phosphorylation of AMP1K and Erk1/2 [70]. Peripherally administered resistin has a much greater effect on the secretion of hormones in the anterior pituitary gland. For example, in sheep, during a long day, the administration of resistin causes an increase in the secretion of LH, FSH, and PRL, while during a short day, the decreased secretion of LH and increased FSH and PRL secretion were found [71].

## 4. Apelin

The first results from research on components of the apelin system (apelin plus its receptor) appeared 30 years ago. In 1993, the apelin receptor (APJ) was cloned and described. Despite the high similarity of the structure of APJ to the angiotensin II receptor, this receptor had no affinity for angiotensin and was initially designated as an orphan receptor [72]. Only five years later, a natural receptor ligand, called apelin, was isolated from the bovine stomach [73]. The apelin gene (*APLN*) encodes an adipokine precursor called preproapelin (made of 77 amino acids), which then undergoes post-translational modifications to proapelin (apelin-55). Mature active forms of apelin, including apelin-36, -17, and -13, and the pyroglutamylated isoform of apelin 13 (pyr-apelin-13), unlike the precursor, are monomers without disulfide bridges between cysteines. The most biologically active form seems to be apelin-13 [73,74,75]. In humans and rats, pyr-apelin-13 is the main isoform in the blood plasma, CNS, and cardiovascular system [76,77,78]. In the lungs and reproductive system of rats, apelin-36 predominates, while in mammary glands, apelin-36 and pyr-apelin-13 are the most abundant forms [79]. The source of many apelin isoforms (e.g., apelin-55, -36, 17, and -13) is the bovine colostrum and human and bovine milk [80,81,82]. Similarly to apelin, apelin eceptors are widely distributed in the CNS and peripheral tissues. The expression of the gene encoding the apelin receptor (*APLNR*) and the presence of the APJ protein have been detected in various structures of the brain, circulatory system, gastrointestinal tract, and reproductive system of humans, mice, and rats [83,84,85,86,87,88]. It is now known that apelin has a number of diverse physiological functions, including an influence on energy homeostasis and the cardiovascular system, and the regulation of the adipoinsular axis with anti-obesity and anti-diabetic properties (for a review, see [89]). Apelin has been classified as an adipokine hormone for quite a long time [90]. The reason for this classification was the detection of apelin mRNA and protein in the adipose tissue of humans, rats, and mice [79,80,85,90,91]. In the white adipose tissue, apelin inhibits adipogenesis of pre-adipocytes and lipolysis in adipocytes; on the other hand, it enhances brown fat adipogenesis and the browning of white adipocytes [92,93]. Apelin protein and *APLN* mRNA were detected in the rat pituitary [74,83,94,95].

Apelin (apelin-17 and apelin-36) was observed in the anterior pituitary cells as well as in the pituitary intermediate lobe (apelin-17) [83] and in the posterior pituitary (apelin-36) [95]. In the pituitary gland of male rats, apelin was detected in the anterior part, mainly in corticotrophs, and to a lesser extent in somatotrophs [83]. In a few cells, the colocalisation of apelin and LH was observed, but FSH-, PRL-, and TSH-immunoreactive cells are devoid of apelin [83]. *APLNR* mRNA has been detected in the pituitary of rats and mice [83,88,94,96]. *APLNR* was highly expressed in the rat anterior and intermediate pituitary [83,96], and it was expressed in lower amounts in the posterior lobe [83]. A high expression of *APLNR* was found in rat corticotrophs [83]. In the mouse pituitary, the expression of this gene and the density of apelin binding sites were high in the anterior part, moderate in the posterior part, and lowest in the intermediate part [88].

Apelin (apelin-17) in an ex vivo perfusion system increased ACTH secretion by rat corticotrophs [83]. In turn, in mice and rats, the i.c.v. administration of pyr-apelin-13 resulted in increases in the ACTH and corticosterone plasma levels, and in rats, it resulted in decreases in the prolactin, FSH, and LH plasma concentrations [97,98]. The authors of both of these publications suggested that the effect of apelin on the secretion of pituitary hormones may be mediated by hypothalamic CRH and AVP. In other studies in rats, a decrease in plasma FSH, LH, and testosterone was also observed after the intraperitoneal administration of apelin-13 [99]. Similar results were obtained in studies on ruminants. In sheep, apelin-13 (administered i.v.) induced significant increases in the concentrations of ACTH, aldosterone, and cortisol in plasma [100].

## 5. Chemerin

The expression of the gene encoding chemerin (*TIG2*, tazarotene-induced gene 2) was initially detected in psoriatic skin lesions in humans [101]. Currently, this gene is also called *RARRES2* (retinoic acid receptor responder 2) [102]. Chemerin, a product of *RARRES2*, was identified in 2003 as an endogenous ligand of chemokine-like receptor 1 (CMKLR1), also called ChemR23 or chemerin receptor 1 [103,104]. Additionally, chemerin is a ligand of two other receptors, GPR1 and CCLR2 [105,106]. Originally, chemerin was described as the factor recruiting leukocytes to inflammatory sites and regulating the immune response and was classified as a chemokine [103]. Later, Goralski et al. [107] observed an unexpectedly high level of chemerin and CMKLR1 expression in human and mouse adipocytes, as well as the regulatory role of the adipokine in adipogenesis, and adipocyte lipid and glucose metabolism. This suggests that chemerin can be classified as a biologically active adipokine. The expressions of *RARRES2* mRNA and chemerin protein are not limited to the skin and adipose tissue. Chemerin expression was also found in the endocrine tissues, gonads, liver, pancreas, and cardiovascular system in humans, rodents, pigs, cows, and turkeys. The expression of chemerin receptors is similarly widespread in the body (for a review, see [102]).

The first report on the expression of chemerin in the pituitary was published by Wittamer et al. [103], where the presence of *RARRES2* and *CMKLR1* mRNA was detected in the human pituitary gland. The chemerin transcript was also detected in the pituitary glands of chimpanzees and baboons [108], and the expression of *RARRES2* and *CMKLR1* was detected in the pituitary glands of mice [109]. Extensive studies of chemerin and its receptors’ gene and protein expression were performed in female pigs. The expression of the chemerin gene and protein was detected in both the anterior and posterior pituitary. The localisation of chemerin protein in various types of pituitary endocrine cells was also examined. Chemerin protein was detected in pig somatotrophs, thyrotropes, and gonadotrophs, but not in corticotrophs [110]. Subsequent studies showed that chemerin receptors (CMKLR1, GPR1, and CCRL2) are present in both the anterior and posterior pituitary of pigs. The presence of the CCRL2 protein was detected in somatotrophs, thyrotrophs, and gonadotrophs, while the CMKLR1 protein was present in thyrotrophs and gonadotrophs. Interestingly, the presence of the GPR1 protein was not demonstrated in any of the types of cells examined, despite the presence of GPR1 in homogenised isolates from the entire pituitary gland [111]. The expression of the chemerin system (chemerin and receptors) in the pig pituitary, and especially in the gonadotrophs, may be related to the regulation of reproductive processes in these animals. It was shown that the abundance of chemerin mRNA transcript and protein in the anterior and posterior lobes changed during the oestrous cycle and early pregnancy, with the highest abundance of chemerin protein in the anterior pituitary on days 2–3 and 10–12 of the oestrous cycle and in days 10–11 of pregnancy. In the case of the posterior pituitary, the relative abundance of chemerin protein was less on days 14–16 compared with other phases of the cycle and was enhanced on days 15–16 of pregnancy [110]. Similarly, the expression of receptor genes and proteins in both parts of the pituitary fluctuates during the oestrous cycle and pregnancy, which could be related to changes in the endocrine status of female pigs [111]. The connection of chemerin with reproductive processes regulated by the pituitary gland (at the pituitary level) is also indicated by the fact that chemerin influenced the basal and stimulated (insulin, GnRH) the secretion of LH and FSH by the isolated pituitary cells. This effect varied depending on the phase of the oestrus cycle when the tissues were collected [110].

## 6. Visfatin

Nicotinamide phosphoribosyltransferase (NAMPT) is a protein that has the activity of both an intracellular enzyme (iNAMPT) and an extracellular cytokine/adipokine (eNAMPT) [112]. The adipokine eNAMPT, originally referred to as pre-B-cell colony-enhancing factor (PBEF), was originally isolated as a presumptive cytokine that enhances the maturation of B-cell precursors [113]. Fukuhara et al. [114] identified PBEF gene mRNA in human visceral fat and named its product visfatin. Visfatin is a 52 kDa protein secreted in mammals not only from adipose tissue [115,116] but also from a number of other tissues, including the central nervous system and gonads [117,118,119]. To date, specific receptors for visfatin have not been definitively identified, and published research results are often contradictory. It has been shown that visfatin can bind to the insulin receptor [114,120] or TLR4 [121]. Additionally, more and more studies indicate that visfatin binds to C-C motif chemokine receptor type 5 [122,123].

Maillard et al. [70] demonstrated visfatin expression in the pituitary glands of female mice and the murine gonadotroph cell line LβT2. Celichowski et al. [124] observed the expression of the visfatin gene in the pituitary glands and isolated corticotrophs of male rats. Furthermore, in male sheep, Dupré et al. [125] showed the expression of visfatin in the intermediate part of the pituitary gland. A dependence of visfatin expression on the stage of the oestrous cycle was also found in the anterior and posterior lobe of the porcine pituitary. The adipokine was present in all types of pituitary endocrine cells, and its secretion was affected by GnRH, FSH, LH, and insulin, depending on the phase cycle; LH stimulated visfatin secretion on days 2–3, and GnRH on days 14–16, whereas FSH and insulin stimulated visfatin secretion on days 17–19 of the cycle, which strongly suggests that visfatin is locally produced in the porcine pituitary in a way that is reliant on the hormonal milieu, which is typical for the reproductive status of pigs [126].

There are few reports regarding the influence of visfatin on endocrine pituitary functions. In vitro studies showed that visfatin inhibited basal LH secretion by the mouse LβT2 cell line [70]. Visfatin stimulated ACTH secretion by the isolated rat corticotrophs, but not by the mouse pituitary corticotroph AtT-20 cell line [124]. The intraperitoneal administration of visfatin enhanced the mRNA abundance of the proopiomelanocortin gene (*POMC*) in the pituitary gland of rats [127]. In turn, in pigs, visfatin administered in vitro affected LH and FSH release and stimulated the pituitary cells’ proliferation in a manner dependent on the phase of the oestrous cycle. The adipokine influenced the cells acting through the insulin receptor and AKT/PI3K, MAPK/ERK1/2, and AMPK signalling pathways [128].

## 7. Irisin

Irisin, named after the Greek messenger goddess Iris, is a novel 12.5-kDa polypeptide hormone with 112 amino acids, which was identified in 2012 by Boström et al. [129]. The adipokine is the product of type 1 membrane protein cleavage encoded by the fibronectin type III domain-containing 5 (*FNDC5*) precursor gene [129]. Until now, no specific receptor for irisin has been identified. The results of recent studies have demonstrated that in fat cells and osteocytes, irisin exerts its action by binding to the members of the αv integrins family, with the highest affinity to αv/β5 integrins. The treatment of osteocytes with irisin significantly stimulated the phosphorylation level of focal adhesion kinase (FAK), the major intracellular signal molecule responsible for integrin signalling. It is also known that irisin treatment in several cell types activates various signalling pathways, including cAMP/PKA, AMPK, Akt/PI3K, MAPK/ERK1/2, p38, and IKK/NF-κB [130,131,132,133,134,135,136].

In addition to skeletal muscle as well as subcutaneous and visceral adipose tissue, irisin is expressed in tissues of the hypothalamic–pituitary–gonadal (HPG) axis, including the rat and tilapia pituitary [134,137,138]. It is suggested that *FNDC5* gene expression can be controlled by tissue- and sex-specific regulatory mechanisms. In monkeys, the *FNDC5* transcript levels were significantly higher in the female muscles, posterior hypothalamus, and whole pituitary than in the corresponding male tissues [139].

Similarly to other adipokines, irisin is likely to have pleiotropic properties. It has been implicated in the regulation of fat and energy metabolism. Adipokine has been suggested to be induced by exercise and can protect against diet-induced obesity, mediated by browning of the white adipose tissue, thus increasing thermogenesis and energy expenditure [129]. Irisin has potential multiple favourable effects on glucose homeostasis and insulin sensitivity by promoting energy expenditure, glucose uptake, and glycogenolysis, as well as by reducing gluconeogenesis, adipogenesis, and lipid accumulation [140]. Many studies have confirmed the anti-inflammatory, anti-apoptotic, anti-oxidative, and pro-angiogenic potential of irisin [141,142,143]. It seems possible that the final effect of irisin action may be partly achieved through a functional relationship with other adipokines [144,145]. Irisin, like other adipokines [17,146,147,148], could be an energy sensor involved in regulating female fertility. The contents of *FNDC5* mRNA and irisin in central and peripheral tissues rise during postnatal development and correlate with the timing of puberty [149].

The findings regarding the effect of irisin on the secretion of pituitary gonadotrophins are contradictory. In an in vivo study, irisin enhanced the LH plasma level and decreased the FSH serum concentration in female rats [150,151]. In other studies on the same model, irisin administered intraperitoneally increased the LH and FSH levels in the blood plasma [152]. Female mice lacking irisin showed decreased levels of LH and FSH in relation to wild-type animals [153]. In turn, in male rats, irisin added i.c.v. decreased the LH and FSH levels [154]. Based on in vitro studies, it was shown that irisin increased the expression of LH and FSH in the pituitary cells by improving the stability of transcription [138]. Moreover, Poretsky et al. [155] showed that irisin infused into the murine pituitary mPit12 cell line stimulates LH production. Irisin showed an opposing influence on gonadotrophin secretion when it was used in combination with GnRH—the adipokine suppressed the stimulatory effect of GnRH in cultures of tilapia pituitary cells [138] and murine mPit12 cell lines [155].

## 8. Conclusions

The influence of adipokines on the selected secretory functions of the pituitary gland is mainly known in rodents, but it is only known to a very limited extent in other species. Their effect on other functions of the pituitary gland, as well as the mechanism of adipokines’ action in gland cells, remains almost completely unknown. The impact of adipokines on the posterior pituitary gland also remains completely unexplored. This is also an indication for further research aimed at completing the missing knowledge about the role of adipokines in the pituitary gland. Nevertheless, the numerous publications cited above indicate that pituitary functions are modified by adipokines, hormones whose main role is to regulate energy homeostasis (Figure 1 and Table 1). Therefore, it seems that the pituitary gland is one of the places where the integration of the regulation of energy and endocrine homeostasis takes place.

## Figures and Tables

**Figure 1 animals-14-00353-f001:**
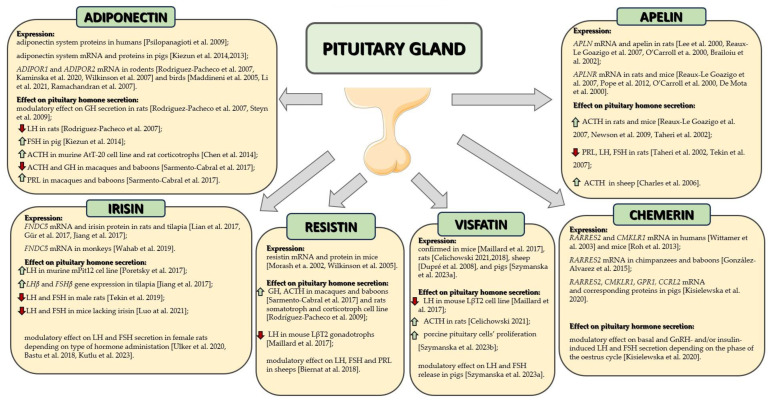
Summary of adipokine expression and action in the pituitary gland. Scheme prepared based on literature data [13,14,15,17,32,37,39,44,45,46,47,48,65,66,69,70,71,74,83,88,94,95,96,97,98,99,100,103,108,109,110,124,125,126,127,128,134,137,138,139,150,151,152,153,154,155]. Arrows in the scheme indicate: ↑—increase, ↓—decrease. This figure was partly generated using Servier Medical Art, provided by Servier, licensed under a Creative Commons Attribution 3.0 unported license.

**Table 1 animals-14-00353-t001:** The impact of adipokines on secretory functions of the pituitary.

Type of Experiment	Adipokine	GH	PRL	ACTH	TSH	LH	FSH	Citation
Rat pituitary primary cell culture	Adiponectin	↓, ↑	↔	↔, ↑	↔	↓	↔	[32,46,47]
	Resistin	↑	ND	ND	ND	ND	ND	[69]
	Visfatin	ND	ND	↑	ND	ND	ND	[124]
Pig pituitary cell culture	Adiponectin	ND	ND	ND	ND	↔	↑	[17]
	Chemerin	ND	ND	ND	ND	↓, ↑#	↑, ↔#	[110]
	Visfatin	ND	ND	ND	ND	↑, ↔#	↓, ↔#	[126]
Baboon and macaque primary pituitary cell culture	Adiponectin	↓	↑	↓	↔	↔	↔	[48]
	Resistin	↑	↔	↑	↔	↔	↔	[48]
Mouse primary pituitary cell culture	Resistin	ND	ND	ND	ND	↓	ND	[70]
	Visfatin	ND	ND	ND	ND	↔	ND	[70]
Cell lines:								
AtT-20	Adiponectin			↑				[47]
	Visfatin			↔				[124]
LβT2	Resistin					↓		[70]
LβT2	Visfatin					↓		[70]
mPit12	Irisin					↑		[155]
Infusion in the following:								
Female sheep	Resistin	ND	↑	ND	ND	↑, ↓ *	↑	[71]
	Apelin	ND	ND	↑	ND	ND	ND	[100]
Female rats	Irisin	ND	ND	ND	ND	↑	↓, ↑	[150,151,152]
Male rats	Apelin	↔	↓	↑	↔	↓	↓	[98,99]
	Irisin	ND	ND	ND	ND	↓	↓	[154]
Male mice	Apelin	ND	ND	↑	ND	ND	ND	[97]

Abbreviation: ↑—increased secretion or plasma concentration; ↓—decreased secretion or plasma concentration; ↔—lack of impact; ND—not determined; *—the effect of adipokine depends on the reproductive stage; #—the effect of adipokine depends on the phase of the oestrous cycle.

## Data Availability

No new data were created in this study. Data sharing is not applicable to this article.

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
