# Peer review of "The Role of Adipokines in the Control of Pituitary Functions"

_animals, 2024, doi:10.3390/ani14020353_

Round 1

Reviewer 1 Report

Comments and Suggestions for Authors

This manuscript describes the role a range of adipokines on pituitary function and the function of each is summarised under its own heading. An important adipokine; leptin is not included even though it does have multiple direct and indirect effects on the pituitary. I suggest it should be included and if not at least some mention in the abstract and introduction as to why it has not been included.  The conclusion is not a conclusion but a general statement that adipokines main role is to regulate energy homeostasis. What is needed here is an attempt to integrate the physiology of all the adipokines into an outcome model of regulation / modulation of pituitary function by adipose tissue, not all of which is about energy homeostasis. Figure 1 attempts to do this but it would be more useful if the key outputs of the pituitary ie LH FSH etc are in the boxes and the effect of the various adipokines is shown having Positive/ negative effects. My specific comments are as follows.

Line 39:  “The pituitary… this sentence should be re written

Line 42: Are all these adipokines expressed in the pituitary? If so, what is their role compared to adipose tissue adipokines.

Line 72: replace functions with physiology

Line 89 “reproductive tract” this is too vague a term break this up into the various organ so the reader knows exactly what you mean.

Line 107-112:  how is T-cadherin a receptor? Perhaps it’s a co-receptor. I am not familiar with topic however this needs more detail here. Co-receptors are well known in other systems.

Line 150; Clarify the status of resistin receptors they are either unknown or known. What does not fully known mean when it appears there is only speculation about receptor identity.

Line 174 Clarify if this is in mice or humans.

Line 221; “Similarly to apelin, “ it is not clear what this means it appears you talking about apelin receptors?

Author Response

Response to Reviewer 1's comments

We would like to thank the Reviewer for the cautious checking of the manuscript and valuable comments, which helped us to enrich the manuscript.

“This manuscript describes the role a range of adipokines on pituitary function and the function of each is summarised under its own heading. An important adipokine; leptin is not included even though it does have multiple direct and indirect effects on the pituitary. I suggest it should be included and if not at least some mention in the abstract and introduction as to why it has not been included.”

We intentionally focused on relatively recently discovered adipokines and their impact on pituitary function. The role of leptin seems to be the best understood of all adipokines. This also applies to the pituitary gland (according to the Pubmed database, 200 articles published in the last 5 years, including 65 reviews). Therefore, it would be difficult to achieve the effect of novelty in the presented work. An appropriate explanation is included in the revised version of the manuscript (lines 62-65). However, we leave the final decision on whether to include a description of the role of leptin in pituitary physiology in the article to the Reviewer and Editor.

“The conclusion is not a conclusion but a general statement that adipokines main role is to regulate energy homeostasis. What is needed here is an attempt to integrate the physiology of all the adipokines into an outcome model of regulation/modulation of pituitary function by adipose tissue, not all of which is about energy homeostasis. Figure 1 attempts to do this but it would be more useful if the key outputs of the pituitary ie LH FSH etc are in the boxes and the effect of the various adipokines is shown having Positive/ negative effects. My specific comments are as follows.”

We agree with the Reviewer. The summary has been modified (lines 393-400). Additionally, we have prepared a table summarizing the influence of adipokines on the secretory functions of the pituitary gland (Table 1)

 Specific comments:

Line 39:  “The pituitary… this sentence should be re written

It has been corrected in the revised version of the manuscript (lines 39-40).

Line 42: Are all these adipokines expressed in the pituitary? If so, what is their role compared to adipose tissue adipokines.

The expression of the described adipokines was detected in the pituitary at the gene or protein level. It should be emphasized, however, that most of the studies concerned rodents and, to a lesser extent, other vertebrates. Details are given in Figure 1.

Line 72: replace functions with physiology

 It has been corrected in the revised version of the manuscript (line 75).

Line 89 “reproductive tract” this is too vague a term break this up into the various organ so the reader knows exactly what you mean.

It has been corrected in the revised version of the manuscript (line 92).

Line 107-112:  how is T-cadherin a receptor? Perhaps it’s a co-receptor. I am not familiar with topic however this needs more detail here. Co-receptors are well known in other systems.

It has been corrected in the revised version of the manuscript (lines 111-112).

Line 150; Clarify the status of resistin receptors they are either unknown or known. What does not fully known mean when it appears there is only speculation about receptor identity.

The status of resistin receptors is not clarified. It is assumed that CAP1 and TLR4 proteins may be such receptors. Direct binding of resistin to CAP1 protein was found in human mononuclear cells (1), while in rat hypothalamus direct binding to TLR4 receptor was described (2). Please, see lines 154-155.

  1. Lee S, Lee H-CH, Kwon Y-W, Lee SE, Cho Y, Kim Y, Lee S, Kim J-Y, Lee J, Yang H-M, Jung I-M, Nam K-Y4, Chung J, Lazar MA, Kim H-S: Adenylyl Cyclase-Associated Protein 1(CAP1) is a Receptor for Human Resistin and Mediates Inflammatory Actions of Human Monocytes. Cell Metab. 2014; 19(3): 484–497. doi:10.1016/j.cmet.2014.01.013
  2. Benomar Y, Gertler A, De Lacy P, Crépin D, Ould Hamouda H, Riffault L, Taouis M: Central Resistin Overexposure Induces Insulin Resistance Through Toll-Like Receptor 4. Diabetes 62:102–114, 2013

Line 174 Clarify if this is in mice or humans.

 It has been corrected in the revised version of the manuscript (lines 174-176).

Line 221; “Similarly to apelin, “ it is not clear what this means it appears you talking about apelin receptors?

It has been corrected in the revised version of the manuscript (line 227).

Reviewer 2 Report

Comments and Suggestions for Authors

This review by Kaminska et al focuses on reviewing particular adipokines and their potential role in regulating pituitary function. There is a lot of valuable information included and the review is well-organized. However, in general, the review reads much like an assemblage of various findings with little to no synthesis of thought as to the actual role of the cytokines in controlling growth, reproduction, or metabolism. What are the gaps in knowledge that still remain? For one, reading through it, it seems like there is very little work that has been done in vivo and much of it has been done in vitro with primary cells or cell lines. This makes determining the actual roles of these peptides very difficult. As another, given the sometimes widespread distribution of the peptides and their receptors, determining the actual importance of pituitary adipokines in development of metabolic issues or reproduction is extremely difficult. My specific comments are below. There are some grammatical issues with the paper and I will try to identify the ones I caught.

L20 – is a key endocrine gland

L25 – a complex of other

L32 – a key endocrine gland

L33 – component of hypothalamus

L39 – to a complex

L43 – are also expressed in the pituitary

L61 – their influence on the pituitary gland are also of importance.

L70 – given the way this paragraph reads, it seems like the focus on function is going to be largely constrained to the anterior pituitary. This also seems to be the focus of the text in the review. Given that, it might be good to state that here.

L88- hypothalamus, pituitary, cardiomyocytes

L124 – in many places, very vague terms are used that do not really help the reader understand the actual role of the peptide. Here, it is mentioned that there are sex-related differences, but no indication of what those actually were.

L129 – In rat pituitary cell

L131 – secretion from isolated

L136 – so it stimulated PRL release in macaques and baboons, but these cells don’t express the receptor in human pituitary? Would that be expected?

L142 – referred to as resistin

L169 – models, production resistin

L179 – on metabolic homeostasis

L187 – so is the resistin found in the pituitary actually made in the pituitary or is the immunopositivity found therein simply resisten transported/released from the arcuate nucleus?

L192 – intraventricularly

L194 – has resisten been localized to any specific pituitary cell types?

L202 – how would resistin have different effects on LH and FSH in different photoperiods (decreased LH but increased FSH during short days?). Also, PRL is typically diametrically opposed to LH and FSH in sheep in its regulation by photoperiod, so it’s odd that it would be regulated in a similar way to LH.

L216 – I would suggest removing the sentence about the isoforms being differentiated. The way the sentence is written, it really does not purvey any real information and is very vague.

L220 – is bovine colostrum….So, I assume this would be an external source of apelin. Why would that be important as the focus here is on pituitary function?  Also, most proteins ingested are simply broken down in the digestive tract, so would this actually have a function in vivo?

L226 – including an influence on energy homeostasis and the cardiovascular system

L242 – one consistent theme that seems to emerge is that the receptor and peptide are often expressed in the pituitary or even within the same cell types. Are autocrine actions of these peptides important?

L265 – expression in human and mouse

L266 – how is this protein involved in metabolism and adipogenesis?

L290 – in what way did it change?

L296 – the expression of many of these proteins change with the estrous cycle. Do any of these genes have steroid response elements in their genes?

L305 – not only from adipose tissue, but from a number of other tissues

L315 – A dependence of visfatin expression on stage of the estrous cycle was also found in the anterior and posterior lobe of the porcine pituitary.

L318 – how was it affected?

L322 – was this just basal LH secretion?

L323 – this is somewhat surprising, what is the sequency homology between the mouse and rat?

L326 – the way this is written, and given that if follows a study where visfatin was given ip, it gives the impression that this was also in vivo. However, it is an in vitro study and should be clearly designated as such.

L333- the product of the type 1

L343 – addition to skeletal muscle

L344 – in tissues of the

L348 – was this whole pituitary?

L351 – suggested to be induced by exercise and protective against diet-induced obesity

L356 – what roles would those be?

L363 – this sentence focuses on puberty, and then the next group of studies were directed toward examining the protein in the context of puberty, but this is entirely unclear with the way it is written and organized. Are there reasons as to why the outcomes were different?

L369 and L374 – I assume “shoved” is supposed to be “showed’?

L377 – The conclusion paragraph is insufficient. As mentioned above, a greater synthesis of thought needs to be relayed heres. What are the knowledge gaps in the work? Where does the future of this work lie?

Comments on the Quality of English Language

Grammatical issues have been addressed in my review

Author Response

Response to Reviewer 2's comments

We would like to thank the Reviewer for all constructive comments and suggestions. Beneath, we respond in detail to each of them.

General comments:

“This review by Kaminska et al focuses on reviewing particular adipokines and their potential role in regulating pituitary function. There is a lot of valuable information included and the review is well-organized. However, in general, the review reads much like an assemblage of various findings with little to no synthesis of thought as to the actual role of the cytokines in controlling growth, reproduction, or metabolism. What are the gaps in knowledge that still remain? For one, reading through it, it seems like there is very little work that has been done in vivo and much of it has been done in vitro with primary cells or cell lines. This makes determining the actual roles of these peptides very difficult. As another, given the sometimes widespread distribution of the peptides and their receptors, determining the actual importance of pituitary adipokines in development of metabolic issues or reproduction is extremely difficult. My specific comments are below. There are some grammatical issues with the paper and I will try to identify the ones I caught.”

We agree with the Reviewer. The summary has been modified (lines 393-400). Additionally, we have prepared a table summarizing the influence of adipokines on the secretory functions of the pituitary gland (Table 1).

Specific comments:

L20 – is a key endocrine gland

It has been corrected in the revised version of the manuscript (line 20).

L25 – a complex of other

It has been corrected in the revised version of the manuscript (line 25).

L32 – a key endocrine gland

It has been corrected in the revised version of the manuscript (line 32).

L33 – component of hypothalamus

It has been corrected in the revised version of the manuscript (line 33).

L39 – to a complex

This sentence was modified (lines 39-40).

L43 – are also expressed in the pituitary

It has been corrected in the revised version of the manuscript (line 43).

L61 – their influence on the pituitary gland are also of importance.

It has been corrected in the revised version of the manuscript (line 61).

L70 – given the way this paragraph reads, it seems like the focus on function is going to be largely constrained to the anterior pituitary. This also seems to be the focus of the text in the review. Given that, it might be good to state that here.

It has been corrected in the revised version of the manuscript (line 74).

L88- hypothalamus, pituitary, cardiomyocytes

It has been corrected in the revised version of the manuscript (line 91).

L124 – in many places, very vague terms are used that do not really help the reader understand the actual role of the peptide. Here, it is mentioned that there are sex-related differences, but no indication of what those actually were.

It has been corrected in the revised version of the manuscript (lines 128-129).

L129 – In rat pituitary cell

It has been corrected in the revised version of the manuscript (line 134).

L131 – secretion from isolated

It has been corrected in the revised version of the manuscript (line 135).

L136 – so it stimulated PRL release in macaques and baboons, but these cells don’t express the receptor in human pituitary? Would that be expected?

The presence of adiponectin receptors in the pituitary of macaques and baboons has not been investigated. Moreover, an indirect effect of adiponectin on PRL secretion from lactotrophic cells in these species via other hormones cannot be ruled out.

L142 – referred to as resistin

It has been corrected in the revised version of the manuscript (line 146).

L169 – models, production resistin

It has been corrected in the revised version of the manuscript (line 174).

L179 – on metabolic homeostasis

It has been corrected in the revised version of the manuscript (line 184).

L187 – so is the resistin found in the pituitary actually made in the pituitary or is the immunopositivity found therein simply resisten transported/released from the arcuate nucleus?

Resistin found in the pituitary gland appears to be of pituitary origin – both resistin mRNA and protein expression were detected in the gland (1). Please, see line 191.

  1. Morash B.A., Willkinson D., Ur E., Wilkinson M. Resistin expression and regulation in mouse pituitary. FEBS Lett. 2002, 526(1-3), 26-30. doi: 10.1016/s0014-5793(02)03108-3.

L192 – intraventricularly

It has been corrected in the revised version of the manuscript (line 197).

L194 – has resisten been localized to any specific pituitary cell types?

Both in the article by Morash et al. (1) cited in the manuscript as well as in a similar work by Lin et al. (2) the occurrence of resistin in individual types of pituitary cells has not been investigated.

  1. Morash B.A., Willkinson D., Ur E., Wilkinson M. Resistin expression and regulation in mouse pituitary. FEBS Lett. 2002, 526(1-3), 26-30. doi: 10.1016/s0014-5793(02)03108-3.
  2. Lin Q, Price SA, Skinner JT, Fan Ch, Yamaji-Kegan K, Johns RA: Systemic evaluation and localization of resistin expression in normal human tissues by a newly developed monoclonal antibody. PLoS ONE 15(7): e0235546. Doi: 10.1371/journal.pone.0235546

L202 – how would resistin have different effects on LH and FSH in different photoperiods (decreased LH but increased FSH during short days?). Also, PRL is typically diametrically opposed to LH and FSH in sheep in its regulation by photoperiod, so it’s odd that it would be regulated in a similar way to LH.

The authors of the cited work explain in the discussion of their article that resistin acts not only on the anterior pituitary but also at the level of the hypothalamus and on GnRH pulse characteristics. They suggest the existence of a paracrine mechanism that operates locally within the pituitary, with PRL being engaged in the paracrine regulation of gonadotrophs’ function – an enhanced PRL concentration in response to resistin could decrease gonadotrophs’ sensitivity to GnRH, resulting in a reduced LH concentration. It is also known that FSH and LH react to GnRH in a different way, depending on the frequency and amplitude of GnRH pulses.

L216 – I would suggest removing the sentence about the isoforms being differentiated. The way the sentence is written, it really does not purvey any real information and is very vague.

The sentence was removed.

L220 – is bovine colostrum….So, I assume this would be an external source of apelin. Why would that be important as the focus here is on pituitary function?  Also, most proteins ingested are simply broken down in the digestive tract, so would this actually have a function in vivo?

In the short period after the calf's birth, colostrum proteins (not only immunoglobulins) are absorbed unchanged. This is possible, among others due to gastric achlorhydria and the presence of a trypsin inhibitor in colostrum.

L226 – including an influence on energy homeostasis and the cardiovascular system

It has been corrected in the revised version of the manuscript (lines 231-234).

L242 – one consistent theme that seems to emerge is that the receptor and peptide are often expressed in the pituitary or even within the same cell types. Are autocrine actions of these peptides important?

In the same cells of male rats (corticotrophs), the presence of apelin protein and APLNR mRNA was found. This does not provide absolute certainty that apelin is produced and that its receptor is present in the same cells. The effect of apelin on the pure fraction of somatotrophs has not been investigated (only on the entire pituitary gland).

L265 – expression in human and mouse

It has been corrected in the revised version of the manuscript (line 271).

L266 – how is this protein involved in metabolism and adipogenesis?

It has been corrected in the revised version of the manuscript (lines 272-274).

L290 – in what way did it change?

It has been corrected in the revised version of the manuscript (lines 299-303).

L296 – the expression of many of these proteins change with the estrous cycle. Do any of these genes have steroid response elements in their genes?

Indeed, most of these genes in several species have oestrogen response elements (1-6). In addition to the ERE-dependent pathway in pituitary cells, there is also an ERE-independent pathway and nongenomic signalling influencing gonadotrophin production (7). What is more, the sensitivity of the gland to steroids, GnRH, and insulin changes throughout the oestrous cycle. The number of gonadotrophs containing oestrogen receptor alpha increases in the follicular phase of the cycle (8). Similarly, the number of receptors for GnRH, insulin, and insulin-like growth factor 1 (IGF-1) is the highest at the end of the cycle (9-12). It is known, moreover, that progesterone is a negative regulator of pituitary GnRH receptors (13, 14), whereas oestradiol increases the receptor expression (15, 16). Oestrogens can also enhance IGF-1 receptors’ concentration in anterior pituitary cells (17) (IGF-1 receptors are able to bind both IGF-1 and insulin). Furthermore, a feedback mechanism between IGF-1 and oestradiol is suggested – oestrogens may sensitise anterior pituitary cells to IGF-1 and IGF-1 can up-regulate oestrogen receptors’ expression (18).

  1. Li G, Tang H, Chen Y, Yin Y, Ogawa S, Liu M, Guo Y, Qi X, Liu Y, Parhar IS, Liu X, Lin H. Estrogen directly stimulates LHb expression at the pituitary level during puberty in female zebrafish. Mol Cell Endocrinol. 2018 Feb 5;461:1-11. doi: 10.1016/j.mce.2017.08.003.
  2. Klungland H, Andersen O, Kisen G, Aleström P, Tora L. Estrogen receptor binds to the salmon GnRH gene in a region with long palindromic sequences. Mol Cell Endocrinol. 1993 Sep;95(1-2):147-54. doi: 10.1016/0303-7207(93)90040-q.
  3. Yaron Z, Gur G, Melamed P, Rosenfeld H, Levavi-Sivan B, Elizur A. Regulation of gonadotropin subunit genes in tilapia. Comp Biochem Physiol B Biochem Mol Biol. 2001 Jun;129(2-3):489-502. doi: 10.1016/s1096-4959(01)00345-1.
  4. Radovick S, Ticknor CM, Nakayama Y, Notides AC, Rahman A, Weintraub BD, Cutler GB Jr, Wondisford FE. Evidence for direct estrogen regulation of the human gonadotropin-releasing hormone gene. J Clin Invest. 1991 Nov;88(5):1649-55. doi: 10.1172/JCI115479.
  5. Christian CA, Glidewell-Kenney C, Jameson JL, Moenter SM. Classical estrogen receptor alpha signaling mediates negative and positive feedback on gonadotropin-releasing hormone neuron firing. Endocrinology. 2008 Nov;149(11):5328-34. doi: 10.1210/en.2008-0520.
  6. Raut S, Kumar AV, Khambata K, Deshpande S, Balasinor NH. Genome-wide identification of estrogen receptor binding sites reveals novel estrogen-responsive pathways in adult male germ cells. Biochem J. 2020 Jun 26;477(12):2115-2131. doi: 10.1042/BCJ20190946.
  7. Glidewell-Kenney C, Weiss J, Hurley LA, Levine JE, Jameson JL. Estrogen receptor alpha signaling pathways differentially regulate gonadotropin subunit gene expression and serum follicle-stimulating hormone in the female mouse. Endocrinology. 2008 Aug;149(8):4168-76. doi: 10.1210/en.2007-1807.
  8. Tobin V.A., Pompolo S., Clarke I.J. The Percentage of Pituitary Gonadotropes with Immunoreactive Oestradiol Receptors Increases in the Follicular Phase of the Ovine Oestrous Cycle. J. Neuroendocr. 2001;13:846–854. doi: 10.1046/j.1365-2826.2001.00701.x.
  9. Clapper J., Taylor A. Components of the Porcine Anterior Pituitary Insulin-like Growth Factor System throughout the Estrous Cycle. Domest. Anim. Endocrinol. 2011;40:67–76. doi: 10.1016/j.domaniend.2010.09.001. 
  10. Luque R.M., Kineman R.D. Impact of Obesity on the Growth Hormone Axis: Evidence for a Direct Inhibitory Effect of Hyperinsulinemia on Pituitary Function. Endocrinology. 2006;147:2754–2763. doi: 10.1210/en.2005-1549.
  11. Navratil A.M., Song H., Hernandez J.B., Cherrington B.D., Santos S.J., Low J.M., Do M.-H.T., Lawson M.A. Insulin Augments Gonadotropin-Releasing Hormone Induction of Translation in LβT2 Cells. Mol. Cell. Endocrinol. 2009;311:47–54. doi: 10.1016/j.mce.2009.07.014.
  12. Buggs C., Weinberg F., Kim E., Wolfe A., Radovick S., Wondisford F. Insulin Augments GnRH-Stimulated LHβ Gene Expression by Egr-1. Mol. Cell. Endocrinol. 2006;249:99–106. doi: 10.1016/j.mce.2006.02.001
  13. Weiss J.M., Polack S., Treeck O., Diedrich K., Ortmann O. Regulation of GnRH I Receptor Gene Expression by the GnRH Agonist Triptorelin, Estradiol, and Progesterone in the Gonadotroph-Derived Cell Line AT3-1. Endocrine. 2006;30:139–144. doi: 10.1385/ENDO:30:1:139.
  14. Cheon M., Park D., Park Y., Kam K., Park S.D., Ryu K. Progesterone Together with Estrogen Attenuates Homologous Upregulation of Gonadotropin-Releasing Hormone Receptor MRNA in Primary Cultured Rat Pituitary Cells. Endocrine. 2000;13:379–384. doi: 10.1385/ENDO:13:3:379.
  15.  Quiñones-Jenab V., Jenab S., Ogawa S., Funabashi T., Weesner G.D., Pfaff D.W. Estrogen Regulation of Gonadotropin-Releasing Hormone Receptor Messenger RNA in Female Rat Pituitary Tissue. Mol. Brain Res. 1996;38:243–250. doi: 10.1016/0169-328X(95)00322-J.
  16. Wu J.C., Sealfon S.C., Miller W.L. Gonadal Hormones and Gonadotropin-Releasing Hormone (GnRH) Alter Messenger Ribonucleic Acid Levels for GnRH Receptors in Sheep. Endocrinology. 1994;134:1846–1850. doi: 10.1210/endo.134.4.8137751.
  17. Rempel L.A., Clapper J.A. Administration of Estradiol-17β Increases Anterior Pituitary IGF-I and Relative Amounts of Serum and Anterior Pituitary IGF-Binding Proteins in Barrows. J. Anim. Sci. 2002;80:214–224. doi: 10.2527/2002.801214x.
  18. Xia Y., Weiss J., Polack S., Diedrich K., Ortmann O. Interactions of Insulin-like Growth Factor-I, Insulin and Estradiol with GnRH-Stimulated Luteinizing Hormone Release from Female Rat Gonadotrophs. Eur. J. Endocrinol. 2001;144:73–79. doi: 10.1530/eje.0.1440073.

L305 – not only from adipose tissue, but from a number of other tissues

It has been corrected in the revised version of the manuscript (line 317).

L315 – A dependence of visfatin expression on stage of the estrous cycle was also found in the anterior and posterior lobe of the porcine pituitary.

It has been corrected in the revised version of the manuscript (lines 327-329).

L318 – how was it affected?

It has been corrected in the revised version of the manuscript (lines 331-333).

L322 – was this just basal LH secretion?

It has been corrected in the revised version of the manuscript (line 336).

L323 – this is somewhat surprising, what is the sequency homology between the mouse and rat?

Sequence homology is 94.1% for mRNA and 99.4% for protein. Perhaps the reason for the observed differences was the use of primary cultures of rat corticotrophs in the first case, and mouse cell lines in the second case.

L326 – the way this is written, and given that if follows a study where visfatin was given ip, it gives the impression that this was also in vivo. However, it is an in vitro study and should be clearly designated as such.

It has been corrected in the revised version of the manuscript (line 340).

L333- the product of the type 1

It has been corrected in the revised version of the manuscript (line 347).

L343 – addition to skeletal muscle

It has been corrected in the revised version of the manuscript (line 357).

L344 – in tissues of the

It has been corrected in the revised version of the manuscript (line 358).

L348 – was this whole pituitary?

It has been corrected in the revised version of the manuscript (line 362).

L351 – suggested to be induced by exercise and protective against diet-induced obesity

It has been corrected in the revised version of the manuscript (lines 365-366).

L356 – what roles would those be?

It has been corrected in the revised version of the manuscript (lines 371-373).

L363 – this sentence focuses on puberty, and then the next group of studies were directed toward examining the protein in the context of puberty, but this is entirely unclear with the way it is written and organized. Are there reasons as to why the outcomes were different?

The sentence was removed (lines 377-379).

L369 and L374 – I assume “shoved” is supposed to be “showed’?

It has been corrected in the revised version of the manuscript (lines 384 and 389).

L377 – The conclusion paragraph is insufficient. As mentioned above, a greater synthesis of thought needs to be relayed heres. What are the knowledge gaps in the work? Where does the future of this work lie?

We agree with the Reviewer. The summary has been modified (lines 393-400). Additionally, we have prepared a table summarizing the influence of adipokines on the secretory functions of the pituitary gland (Table 1).

Round 2

Reviewer 1 Report

Comments and Suggestions for Authors

I agree there is no need to include leptin in the review but it should be mentioned and this has been done.